# Clinical Experience with Genome-Wide Noninvasive Prenatal Screening in a Large Cohort of Twin Pregnancies

**DOI:** 10.3390/genes14050982

**Published:** 2023-04-26

**Authors:** Luigia De Falco, Giovanni Savarese, Pasquale Savarese, Nadia Petrillo, Monica Ianniello, Raffaella Ruggiero, Teresa Suero, Cosimo Barbato, Alessio Mori, Cristina Ramiro, Luigi Della Corte, Gabriele Saccone, Attilio Di Spiezio Sardo, Antonio Fico

**Affiliations:** 1AMES, Centro Polidiagnostico Strumentale, 80013 Naples, Italy; giovanni.savarese@centroames.it (G.S.); pasquale.savarese82@gmail.com (P.S.); nadia.petrillo@centroames.it (N.P.); monica.ianniello@centroames.it (M.I.); cosimo_barbato@icloud.com (C.B.); mrolssa@gmail.com (A.M.); centroames@libero.it (A.F.); 2Fondazione Genetica per la Vita Onlus, 80132 Naples, Italy; 3Department of Neuroscience, Reproductive Sciences and Dentistry, School of Medicine, University of Naples Federico II, 80013 Naples, Italy; dellacorte.luigi25@gmail.com (L.D.C.); gabriele.saccone.1990@gmail.com (G.S.); 4Department of Public Health, Gynecology Unit—Federico II University Hospital of Naples, 80138 Naples, Italy; attiliodispiezio@libero.it

**Keywords:** non-invasive prenatal testing, genome-wide sequencing, twin pregnancies, failure rate, sensitivity, specificity

## Abstract

Non-invasive prenatal screening (NIPS) in twin gestations has been shown to have high detection rates and low false-positive rates for trisomy 21, as seen in singleton pregnancies, although there have been few large cohort twin studies, genome-wide studies in particular, to date. In this study, we looked at the performance of genome-wide NIPT in a large cohort consisting of 1244 twin pregnancy samples collected over a two-year period in a single laboratory in Italy. All samples underwent an NIPS for common trisomies, with 61.5% of study participants choosing to undergo genome-wide NIPS for additional fetal anomalies (namely, rare autosomal aneuploidies and CNVs). There were nine initial no-call results, all of which were resolved upon retest. Based on our NIPS results, 17 samples were at high risk for trisomy 21, one for trisomy 18, six for a rare autosomal aneuploidy, and four for a CNV. Clinical follow-up was available for 27 out of 29 high-risk cases; a sensitivity of 100%, a specificity of 99.9%, and a PPV of 94.4% were noted for trisomy 21. Clinical follow-up was also available for 1110 (96.6%) of the low-risk cases, all of which were true negatives. In conclusion, we found that NIPS was a reliable screening approach for trisomy 21 in twin pregnancies.

## 1. Introduction

The rates of twin pregnancies have increased over the last four decades in many countries, likely due to several factors, including increased maternal age at birth and the increased use of assisted reproductive techniques [1,2]. On average, about two thirds of twin pregnancies are dizygotic, and one third are monozygotic [3]. With respect to chorionicity, all dizygotic pregnancies and about one third of monozygotic pregnancies are dichorionic [3]. Both zygosity and chorionicity have been shown to vary by maternal age [4]. Chorionicity should be determined using ultrasound in the first trimester, if possible, as chorionicity in twin pregnancies can impact the screening process and is also strongly associated with pregnancy-related risks [3,4,5,6].

There are several different approaches that can be used for traditional screening for aneuploidy in twins. In the first trimester, nuchal translucency (NT) screening combined with maternal age can be used, with reported detection rates for trisomy 21 ranging from 69–80% at a 5% false-positive rate [7]. First-trimester combined screening (measurement of NT and maternal serum markers) for trisomy 21 in twin pregnancies can also be used and has been shown to have a detection rate of 89.3%, with a false-positive rate of 5.4% [8]. Another approach is the use of integrated screening represented by NT measurement first plus a second trimester serum screening [3]. It is typically recommended that high-risk screening results are confirmed though the use of invasive diagnostic techniques such as chorionic villus sampling (CVS) and amniocentesis. Moreover, if we choose CVS over amniocentesis, it is important that both layers of the placenta are analyzed, as this would give the most accurate risk assessment for fetal involvement. It should be noted that the amniocentesis procedure may be associated with a pregnancy loss rate of 0.3–2.2% in twin pregnancies [7], whilst the use of CVS has been found to have technical challenges and a high risk of sampling error when performed in twin pregnancies [3]. Non-invasive prenatal testing (NIPS) using cell-free DNA in maternal blood for common fetal aneuploidies was introduced commercially in 2011 following the discovery of circulating feto-placental-derived cell-free DNA in maternal plasma in 1997 [9]. NIPS for multifetal gestations became clinically available in 2012 [10]. The list of conditions that NIPS can screen for has expanded over the years, with screening for genome-wide fetal chromosomal anomalies such as rare autosomal anomalies (RAAs) and copy number variants (CNVs) being now an option that is available for many patients. However, the positive predictive values (PPVs) for RAA and CNV conditions are significantly lower than the PPVs for common aneuploidies, and large-scale outcome studies have not been performed [11,12]. 

A recent study by the Dutch NIPT Consortium found that an additional finding was reported for about one in every 275 women that underwent genome-wide NIPS, with most of these additional findings having a clinical impact [13]. Currently, several professional societies are supportive of the use of NIPS in twin pregnancies, with some restrictions [1,14,15,16].

Currently only limited data are available on the performance of NIPS, genome-wide NIPS in particular, in multifetal gestations. The main aim of this study was to determine the performance of genome-wide NIPS for fetal anomalies in a large cohort of twin pregnancy samples. Our study found that NIPS was an effective screening approach for trisomy 21 in twin pregnancies, with an observed sensitivity and specificity of 100% and 99.9%, respectively, as well as a PPV of 94.4% and no reported false negatives.

## 2. Materials and Methods

Samples were collected from patients with twin pregnancies referred to the AMES laboratory in Naples for genome-wide cell-free (cf) DNA screening between March 2020 and December 2022. The laboratory is accredited (UNI EN ISO 9001:2008) for prenatal testing and genetic testing. The study population included both low-risk women choosing NIPS as a first-tier test as well as women considered to be at elevated risk for fetal aneuploidies based on advanced maternal age, a prior affected pregnancy, positive first trimester combined test results (risk above 1/270 or 1/300, depending on individual hospital criteria), abnormal ultrasound findings (including a nuchal translucency of >3.5 mm), parents with balanced chromosomal abnormalities or other chromosomal rearrangements of a sufficient size to be detectable by our assay, or a family history of aneuploidy.

All patients had to have a maternal age of at least 18 years, with a minimum gestational age (GA) of 10 weeks for inclusion in the study. The cohort included monochorionic and dichorionic twin pregnancies and triplet pregnancies both naturally conceived or conceived with assisted reproductive techniques (ART). Chorionicity was assessed by ultrasound and different sexes. All patients received pre-test counselling, and written informed consent was obtained before blood collection. Patients were offered the choice of screening for common trisomies only or to undergo genome-wide NIPT, which included screening for common trisomies, RAAs, and CNVs; presence or absence of chromosome Y was also reported for all patients. In the case of an abnormal test result, additional counselling was provided by a clinical geneticist or obstetrician [17], and confirmatory testing in material obtained via amniocentesis or chorionic villus sampling was offered.

Genome-wide cfDNA screening was carried out using a VeriSeq^TM^ NIPT Solution v2 assay (Illumina, Inc., San Diego, CA, USA). Samples were collected and processed for cfDNA screening, similar to as previously described [18]. Briefly, cfDNA was extracted from samples that were either collected at the laboratory or sent to the laboratory according to the VeriSeq NIPT Solution v2 package insert [19,20,21]. In accordance with the laboratory routine, the assays were run as singleton mode. Whole-genome sequencing was carried out using the NextSeq™ 550Dx sequencer (Illumina, Inc.), followed by bioinformatic analysis with the VeriSeq NIPT Solution Software v2 (https://emea.support.illumina.com/content/dam/illumina-support/documents/documentation/chemistry_documentation/veriseq-nipt-v2/veriseq-nipt-solution-v2-software-guide-ivd-1000000067940-06.pdf, accessed on 1 December 2022), similar to previously published [20]. Sample results were classified using the VeriSeq NIPT Solution v2 Assay Software and analysis of “raw data”, as reported previously [18,21]. Samples failed when the sequencing coverage was judged insufficient based on the fetal fraction (FF) estimate for the sample, as indicated by the Individualized Fetal Fraction Confidence Test (iFACT), which is a quality control parameter of the VeriSeq NIPT Solution v2 assay [20].

Follow-up diagnostic testing was performed for all cases with high-risk NIPS results, most of which were carried out in the AMES laboratories at no additional cost. In cases of low-risk NIPS results, data on pregnancy outcomes were provided by the treating physician or by the patient. Follow-up data were entered into a database. Clinical truth was determined via invasive testing (chorionic villus sampling or amniocentesis), products of conception (POC) testing, karyotype at birth, and newborn physical exam. Chromosome analysis included the use of karyotyping and Single Nucleotide Polymorphism (SNP) testing, by use of the Human-CytoSNP-12 v2.1 BeadChip, which also allowed testing for the presence of a uniparental disomy when applicable (nonverified trisomy of chromosome 6, 7, 11, 14, 15, or 20). For karyotype analysis, amniotic fluid was drawn, and genomic DNA was extracted from the amniocyte using the QIAamp DNA Blood Mini Kit (Qiagen). Aneuploidies for chromosomes 13, 18, and 21 and the sex chromosomes were first screened by a quantitative fluorescence polymerase chain reaction (QF-PCR; Devyser Compact v3, Devyser), as previously described [22]. The amplified DNA samples were separated by electrophoresis using an ABI 3130xl Genetic Analyzer, and the analysis of each allele for specific markers was performed using the GeneMapper Software ver. 4.0 (Applied Biosystems, Waltham, MA, USA). GTG-banding analysis of amniotic fluid was performed in established cell culture following standard laboratory protocols. Metaphases were analyzed with the CytoVision software (CytoVision, AB Imaging). SNP array analysis was performed using a HumanCytoSNP-12 v2.1 kit, following the manufacturer’s protocol (http://www.support.illumina.com/array/protocols.ilmn, accessed on 1 December 2022). Stained Bead-Chips were scanned with a HiScan™SQ System (Illumina, Inc.). Data were generated with GenomeStudio software (Illumina, Inc.) and analyzed with the Bluefuse Multi Software (Illumina, Inc.). All CNVs > 100 Kb were interrogated. All results were reported according to the GRCh37 (hg19) assembly.

Statistical analysis included the calculation of positive predictive values, as well as sensitivity, specificity, and their respective 2-sided 95% confidence intervals (CIs). CIs were based on Wilson’s score method [23]. Statistical analysis of the data was conducted utilizing the statistical set SPSS for Windows (version 20 SPSS Inc., Chicago, IL, USA).

The study was conducted according to the guidelines of the Declaration of Helsinki. This study was approved by the Ethics Committee of the University of Naples Federico II (protocol number 219/19).

## 3. Results

### 3.1. Details of Study Cohort

A total of 1254 samples were collected, processed, and analyzed for inclusion in the study, namely, 1244 twin samples and 10 triplet samples (Figure 1). The ten patients with triplet pregnancies had an average maternal age of 36.6 years and an average gestational age of 12.24 weeks (Appendix A). Of these, one received a high-risk NIPS result for trisomy 21, with follow-up analysis showing that this was a true positive. The remaining nine samples received a NIPS result of “no fetal anomaly detected”, and all were confirmed as true negatives. These ten triplet samples were not included in the final study cohort. Demographics for the 1244 patients with twin pregnancies are shown in Table 1. NIPS indications were available for 955 (76.8%) patients, with the majority of these listing advanced maternal age as the indication for screening (50.5%). The mean maternal age range was 18–48 years. The vast majority of patients (92%) in our study cohort underwent NIPT in the first trimester of pregnancy; the mean and median gestational ages were 12 weeks.

### 3.2. cfDNA Screening Results for Twin Samples

Of the 1244 twin samples that underwent NIPS, a total of nine failed to give a result upon first NIPS analysis (eight failed iFACT and one had data outside the expected range). The average gestational age for these nine samples was 11.54 weeks and the average fetal fraction was 18%. A result was obtained for six of the nine samples after a second NIPT analysis was carried out on the initial blood sample, and the other three samples were resolved following a second blood draw. Therefore, a result was obtained for all of the twin samples (*n* = 1244) in the study cohort. Of these 1244 final study samples, 66 (5.2%) were monochorionic and 1080 (86.8%) were dichorionic; chorionicity was unknown for the remaining 98 (7.8%) samples (Table 1). The average fetal fraction for all samples in the study cohort was 12.4%, with an average of 12.67% for monochorionic pregnancies and an average of 12.15% for dichorionic pregnancies (Figure 2).

All the twin samples underwent NIPS for common trisomies, with 18 samples found to be high risk for trisomy 21 and one for trisomy 18 (Table 2). In addition, 61.5% of study participants (765/1244) chose to undergo genome-wide NIPS. Of these, there were six that were at high risk for the presence of a RAA and four that were at high risk for the presence of a CNV by NIPS. All the high-risk NIPS samples were from dichorionic twin pregnancies.

### 3.3. Clinical Outcomes for High-Risk NIPT Cases

Clinical and diagnostic follow-up was available for 27 (93.10%) of the 29 cases with a high-risk result following NIPS, see Table 2. Of the 18 women with trisomy 21, 17 were true positives (Table 2), and one was a discordant result, with a positive predictive value (PPV) of 94.4%. Sensitivity and specificity for trisomy 21 were 100% (17/17; 95% CI: 81.57–100%) and 99.9% (1109/1110; 95% CI: 99.54–100%), respectively. In the one discordant trisomy 21 case, the maternal age was 40 years, the gestational age at time of NIPS blood draw was 11 + 0 weeks, and the FF was 5%. Intrauterine fetal demise (IUFD) of one twin was noted at 13 weeks of GA (after the NIPS analysis but prior to diagnostic testing), and no confirmatory testing was carried out on the demised twin. Confirmatory diagnostic testing was carried out using amniocentesis on the surviving twin at 15 + 5 weeks GA, with normal results. It is very likely that positive NIPS reflected the result of the demised co-twin. The one trisomy 18 case was confirmed as a true positive following karyotype analysis on products on conception; the affected co-twin was spontaneously aborted at 27 weeks of gestation and both twins were simultaneously delivered at 38 + 3 weeks. During the first trimester of pregnancy, nuchal translucency was performed, and a cystic igroma for one fetus was found. The dichorionic pregnancy was a result of an egg donation. Sensitivities and specificities for trisomy 18 were 100% (1/1; 95% CI: 20.65–100%) and 100% (1076/1076; 95% CI: 99.61–100%), respectively. Of the ten cases that were at high risk for additional fetal anomalies (i.e., RAAs and CNVs), five were found to be discordant with the NIPS result, three were lost to follow-up, and the other two had no confirmatory diagnostic testing. Clinical follow-up was also available for 1110 (89.2%) of the cases that were reported by NIPS as “no fetal anomaly detected”, and all of these were found to be true negatives.

Further details of the six RAA and four CNV cases are shown in Table 3, including pregnancy outcomes where available. None of the ten cases had any ultrasound anomalies. In case #6, a trisomy 9 was noted on NIPS. The patient underwent a spontaneous miscarriage at 12 + 5 weeks GA, and a follow-up NIPS at 15 + 6 weeks GA did not show presence of a trisomy 9 (a low-risk result was obtained). No diagnostic testing was performed on either the demised twin or the surviving twin, and the patient went on to deliver a healthy baby. One of the RAA cases (case #3, trisomy 5) was lost to follow-up, and the other three RAA cases (two trisomy 3 cases and one trisomy 8 case) were found to be false positives following fetal testing. For the trisomy 8 case (case #5), a high mosaic ratio (5.48) and high mosaic log likelihood ratio trisomy score (7766) were noted on the NIPS report. This dichorionic pregnancy was a result of a natural conception, and the pregnant woman referred to a previous miscarriage and a previous pregnancy with a delivery of a healthy baby. In addition, both parents had a normal standard karyotype. Ultrasounds at 13 and 19 weeks of gestation showed no malformations. For the four CNV cases, two were found to be false positives following amniocentesis and the other case was lost to follow-up. No placental testing was carried out for any of the RAA or CNV cases with follow-up.

## 4. Discussion

Non-invasive prenatal testing using cfDNA is typically carried out to screen for common fetal chromosomal anomalies, with the option to screen for a wider range of chromosomal changes (expanded/genome-wide NIPS) becoming increasingly available. However, studies that investigated the clinical relevance of these additional findings from whole-genome sequencing-based NIPT are limited. A recent publication showed that genome-wide aneuploidy screening for the presence of rare autosomal aneuploidies could be beneficial in a number of clinical situations, such as providing a possible explanation for an adverse pregnancy outcome or resulting in a change in pregnancy management [24].

Our study was carried out to determine the performance of genome-wide NIPS in a large cohort of twin pregnancies. An initial no-call rate of 0.87% was reported, with all of the failed samples resolving upon repeat NIPS analysis. Previous studies, where NIPS was carried out in twin samples, found initial no-call rates ranging from 1.6 to 13.2%, with a median of 3.6% [1], whilst a recent study by Khalil et al. noted a low failure rate of 0.31% [25]. In our study, the use of iFACT, which employed a combined threshold for sample calling (using sequencing coverage and FF), allowed informative calls to be made on samples with low FFs. Furthermore, with sample reprocessing (without redraw), coverage could change, allowing some failed samples to be rescued and avoiding a no-call result that would require the patient to undergo a repeat blood draw, leading to a delay in results or an invasive diagnostic procedure. Our study also had a screen-positive rate of 2.3% for genome-wide fetal anomalies, with all of the affected cases being from dichorionic twin pregnancies. In addition, we found that NIPS performed accurately in our ten triplet gestations.

The majority of high-risk calls in our final study cohort were trisomy 21 cases, which had an observed sensitivity of 100% and specificity of 99.9%. The sensitivity for trisomy 21 in our study was higher than the estimated sensitivity (96.4%) for trisomy 21 in twin samples provided in the assay package insert [19], whilst the specificity was the same (99.9%). A recent meta-analysis of twin pregnancies by Judah et al. [26] found that NIPS was able to detect trisomy 21 with a detection rate of 99% and a false-positive rate of <0.02%, which was similar to that seen in singleton pregnancies [27]. We also reported a PPV of 94.4% for trisomy 21, which was higher than that observed in a recent study by Chibuk et al. that noted a PPV for trisomy 21 of 78.7% in twin pregnancy samples [28]. In addition, clinical outcomes were available for most of our low-risk cases, and no false negatives were reported. As noted above, over 90% of our cases were from dichorionic twin pregnancies. In dichorionic pregnancies with one affected twin, false negatives may occur if there is a disparity in the amount of fetal fraction contributed by each fetus in a dizygotic pair [7]. This is particularly true for cases of trisomy 18 or trisomy 13, where the fetal fraction for the affected twin will be even lower due to the presence of less placental tissue [6]. There were no trisomy 13 samples in our cohort and only one trisomy 18 sample.

Few studies have looked at the performance of genome-wide NIPS in a large cohort of twin samples. A recent study by van Riel et al. [29] that carried out genome-wide NIPSs in a large cohort of multiple gestations noted the presence of an RAA in thirteen of the twin samples in their study cohort (not including the vanishing twin gestations). Of these, five were lost to follow-up, and the other eight samples were found to be discordant following amniocentesis. As detailed above, there were a total of six samples in our study where the NIPS result was discordant with the diagnostic testing. One of these was a trisomy 21 case where the patient in question underwent a fetal demise following NIPS but prior to confirmatory diagnostic testing. However, no POC analysis was carried out on the demised twin, with confirmatory testing being carried out on the surviving twin only. It is, therefore, possible that the demised twin was the affected twin, which would explain the discordance between the NIPS result and the clinical outcome. Similarly, in the discordant trisomy 9 case in our study, the patient experienced a co-twin demise during the pregnancy, and no POC analysis was carried out. However, a repeat NIPS at a later gestational age returned a low-risk result, again suggesting that the demised twin was affected. Similar results were noted in a recent study by Mossfield et al. [24], in which four cases of a co-twin demise returned normal NIPS results upon a repeat blood draw carried out at a later gestational age. In that study, the authors suggested that the RAA result on the NIPS may have been attributed to the demised twin and that this may provide an explanation to the patient for the loss of that twin. The other three RAA samples in our study that had clinical follow-up were found to be discordant with the NIPS result following fetal testing (either amniocentesis or postnatal testing). However, none of these cases underwent placental testing. For one of these cases (trisomy 8), a high mosaic ratio and high mosaic log likelihood ratio trisomy score were noted on the NIPS report. Trisomy 8 could also have occurred as a low-level mosaic in the mother, but, thus far, she is in good health. However, without additional information, we are unable to provide an explanation for this discordant NIPS result. Some recent publications looking at genome-wide NIPS have noted that concordance of RAA results can be based on either fetal or placental testing or both [13,24]. This is mainly due to the fact that RAAs are often found to be present in placental tissues only, i.e., confined placental mosaicism (CPM). It is, therefore, possible that the RAA noted on NIPS was present in the placenta in one or more of our cases. Cases with CPM of RAAs have been shown to be associated with pregnancy complications and adverse pregnancy outcomes including fetal growth restriction, pre-eclampsia, and preterm birth [13,24,30], highlighting the importance of screening for these additional anomalies. Similarly, the two CNV cases in our cohort with clinical follow-ups also only underwent diagnostic testing of fetal but not placental tissues, with both cases found to be discordant.

This study had several strengths: firstly, the large number of twin samples in our cohort; in addition, we had clinical follow-up on 96.5% of all the analyzed samples, including the majority of cases reported by NIPS as “no fetal anomaly detected”. A limitation of our study was the low number of affected twin samples, with a NIPS screen-positive rate of 2.3% for all samples and only nine cases that were at high risk for additional fetal anomalies (i.e., RAAs and CNVs). However, due to the low prevalence of these additional fetal anomalies, this will be a limitation of all studies looking at genome-wide NIPT in pregnant populations. The number of affected cases in our study were too low to make any definitive conclusions about the performance of NIPS for fetal anomalies other than trisomy 21. In this context, a follow-up would be interesting, which, also, considers how parents cope with the high rate of false positives with RAA and CNVs.

Finally, as noted above, no placental testing was carried out for the cases with a high-risk NIPS result for a rare autosomal aneuploidy or CNV; thus, a biological etiology such as CPM could not be ruled out.

In conclusion, our study found that NIPS was an effective and reliable screening approach for trisomy 21 in twin pregnancies, with a PPV of 94.4%. Future studies with additional cases of RAAs and CNVs in twins, along with follow-up fetal and placental diagnostic testing, will help to further evaluate the performance of genome-wide NIPS in twin gestations.

## Figures and Tables

**Figure 1 genes-14-00982-f001:**
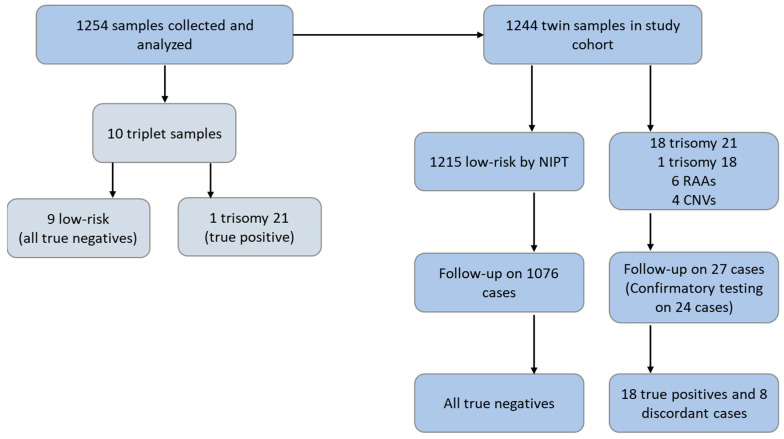
Study flow chart.

**Figure 2 genes-14-00982-f002:**
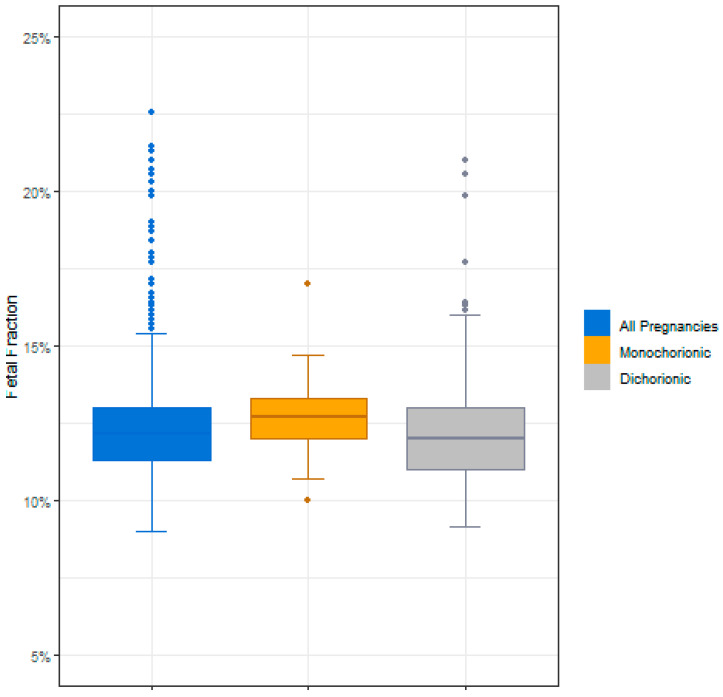
Fetal fractions for all samples, monochorionic samples, and dichorionic samples.

**Table 1 genes-14-00982-t001:** Patient characteristics of the study cohort (*n* = 1244).

Characteristic	Value
Indications for NIPT, *n* (%) Advanced maternal age Based on traditional screening ^1^ Based on pregnancy history ^2^ Parental carrier of balanced translocation ^3^ Patient preference Unknown	628 (50.5) 25 (2.0) 12 (1.0) 2 (0.2) 288 (23.1) 289 (23.2)
Maternal age/Egg donor age, y Mean Median IQR	35.22 35.05 6.48
Gestational age at NIPT, days Mean Median IQR	12.4 12.14 1.71
Trimester of screening, *n* (%) First (up to 14 wk) Second (14–27 wk)	1111(89.3) 133 (10.7)
ART pregnancies, *n* (%) Total IVF ICSI IUI Egg donation ^4^ PGT-A transfer euploid embryo Not specified	370 (30.5) 12 (1.0) 15 (1.3) 5 (0.5) 35 (2.6) 7 (0.6) 282 (24.5)
Chorionicity, *n* (%) Dichorionic pregnancy ^5^ Monochorionic pregnancy ^6^ Chorionicity unknown	1080 (86.8) 66 5.3) 98 (7.8)

ART, assisted reproductive techniques; ICSI, intracytoplasmic sperm injection; IQR, interquartile range; IUI, intrauterine insemination; IVF, in vitro fertilization; *n*, number; NIPT, noninvasive prenatal testing; PGT-A, preimplantation genetic testing for aneuploidy; y, year. ^1^ FCT-serum NT-2nd trimester serum; includes ultrasound findings such as renal pyelectasis and cystic hygroma. ^2^ Previous pregnancies with aneuploidy.^3^ Includes parents with balanced chromosomal abnormalities or other chromosomal rearrangements of sufficient size to be detectable by our assay. ^4^ Two sample included both oocyte and sperm donation. ^5^ Based on 1st trimester ultrasound, different sex, or postnatal placental examination. ^6^ Based on 1st trimester ultrasound or postnatal placental examination.

**Table 2 genes-14-00982-t002:** Clinical outcomes for high-risk NIPT cases.

**NIPT Result**	** *n* **	**Lost to ** **Follow-Up**	**No Confirmatory Testing**	**True ** **Positives**	**Discordant Result**
Trisomy 21	18	0	1 *	17	1
Trisomy 18	1	0	0	1	0
RAA	6	1	1 *	0	4
CNV	4	1	0	0	2

CNV, copy number variant; RAA, rare autosomal aneuploidy. * Patient experienced an intrauterine fetal demise of one twin.

**Table 3 genes-14-00982-t003:** Characteristics of cases and pregnancy outcomes or patients with a rare autosomal aneuploidy or CNV identified by NIPT (*n* = 10).

Case	Indication for NIPT	GA at Blood Draw, wk	NIPT Result	FFE, %	Confirmatory Testing	Pregnancy Outcomes
RAA Cases						
1	AMA	10	Trisomy 3	9	Amnio; RAA not confirmed	Two healthy babies
2	Patient preference	13 + 6	Trisomy 3	12	Amnio; RAA not confirmed	Two healthy babies
3	AMA	15 + 2	Trisomy 5	11	------	------
4	AMA	13 + 1	Trisomy 5	7	None	Two healthy babies
5	AMA	11 + 4	Trisomy 8	13	Postnatal karyotype; RAA not confirmed	Two healthy babies
6	Patient preference	11 + 5	Trisomy 9	10	None *	One healthy baby
CNV Cases						
7	Patient preference	12 + 6	dup(9)(p24.3p13.1)	15	-----	-----
8	AMA	12 + 6	dup(18)(p11.31p11.1)	20	Amnio; CNV not confirmed	Two healthy babies
9	AMA	11 + 4	del(1)(p36.32p32.3)(15)	9	Amnio; CNV not confirmed	Two healthy babies
10	AMA	13	dup(7)(p15.2p14.2)	8	None	Two healthy babies

AMA, advanced maternal age; Amnio, amniocentesis; CNV, copy number variant; FFE, fetal fraction estimate; MA, maternal age; RAA, rare autosomal aneuploidy; wk, week; yr, year. * Patient experienced a co-twin demise at 12 + 5 weeks GA. A second blood draw and NIPT analysis was carried out at 15 + 6 weeks GA, and a low-risk result was obtained; no diagnostic testing was carried out on either the demised twin or the surviving twin.

## Data Availability

Protocols and deidentified, aggregated data that underlie the results reported in this article are available for non-commercial scientific purposes upon reasonable request from the corresponding author.

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
