# Peer review of "Clinical Experience with Genome-Wide Noninvasive Prenatal Screening in a Large Cohort of Twin Pregnancies"

_genes, 2023, doi:10.3390/genes14050982_

Round 1

Reviewer 1 Report

De Falco and colleagues report a large NIPT study in twin pregnancies from an Italian cohort.

The authors found similar sensitivities, specificities and PPVs in twin pregnancies as for singletons even better values but this might be due to the overall low number of positives in the cohort as discussed. There are only some minor points to be addressed:

-          In the introduction the statement ‘The list of conditions that NIPT can screen for has expanded over the years, with screening for genome-wide fetal chromosomal anomalies such as rare autosomal anomalies (RAAs) and copy number variants (CNVs) now an option that is available for many patients.’ is clearly too positive and sounds like an uncritical test for any of the abnormalities! Later in the manuscript the authors speak in more relative terms as even their own data (and large other studies) are critical in NIPT testing for other abnormalities than trisomy 21!

-          The global statement ‘It is typically recommended that high-risk screening results are confirmed though the use of invasive diagnostic techniques such as chorionic villus sampling (CVS) and amniocentesis.‘ is wrong. As with CVS a CPM can be confirmed with no true result for the fetus – be more precise.

-          Please provide the maternal age range.

-          Figure 1 indicates 18 trisomy 21 cases this is in contrast to text line 1991 ‘All the twin samples underwent NIPT for common trisomies, with 17 samples found to be high-risk for trisomy 21’?!

-          As the authors discuss the number of affected twins other than trisomy 21 like trisomy 18 and even trisomy 13 were even not seen. Other RAAs and CNVs had a very high rate of false positives. Therefore, a follow up would be interesting of how parents cope with these test results and with the knowledge if they would have had refused a NIPT test other than for trisomy 21?

Author Response

De Falco and colleagues report a large NIPT study in twin pregnancies from an Italian cohort.

The authors found similar sensitivities, specificities and PPVs in twin pregnancies as for singletons even better values but this might be due to the overall low number of positives in the cohort as discussed. There are only some minor points to be addressed:

-          In the introduction the statement ‘The list of conditions that NIPT can screen for has expanded over the years, with screening for genome-wide fetal chromosomal anomalies such as rare autosomal anomalies (RAAs) and copy number variants (CNVs) now an option that is available for many patients.’ is clearly too positive and sounds like an uncritical test for any of the abnormalities! Later in the manuscript the authors speak in more relative terms as even their own data (and large other studies) are critical in NIPT testing for other abnormalities than trisomy 21!

      Response: Thanks for your comment. We added the sentence “However, the positive predictive values (PPVs) for RAA and CNVs conditions are significantly lower than the PPVs for common aneuploidies and large-scale outcome studies have not been performed Lane 63-65

-          The global statement ‘It is typically recommended that high-risk screening results are confirmed though the use of invasive diagnostic techniques such as chorionic villus sampling (CVS) and amniocentesis. ‘ is wrong. As with CVS a CPM can be confirmed with no true result for the fetus – be more precise.

      Response: Thanks for your comment. As the reviewer suggested we added the sentence: “Moreover, if we choose CVS over amniocentesis, it is important that both layers of the placenta are analyzed, as this would give the most accurate risk assessment for fetal involvement” Lane 49-51.

-          Please provide the maternal age range.

       Response: Thanks for your comment. As the reviewer suggested, we added the maternal age range (18-48). Lane 161

-          Figure 1 indicates 18 trisomy 21 cases this is in contrast to text line 1991 ‘All the twin samples underwent NIPT for common trisomies, with 17 samples found to be high-risk for trisomy 21’?!

      Response: Thanks for your comment. We replaced 17 with 18 at lane 191 (now lane 197).

-          As the authors discuss the number of affected twins other than trisomy 21 like trisomy 18 and even trisomy 13 were even not seen. Other RAAs and CNVs had a very high rate of false positives. Therefore, a follow up would be interesting of how parents cope with these test results and with the knowledge if they would have had refused a NIPT test other than for trisomy 21?

    Response: Thanks for your comment. We agree with this reviewer that a follow up would be interesting of how parents cope with the high rate of false positive results of RAA and CNVs and with the knowledge if they would have had refused a NIPT test other than for trisomy 21. Morevover, before performing NIPS analysis, a pre-test counselling was performed by a genetic counselor about the limits of NIPS expanded. As the reviewer suggested, we added a sentence “In this context, a follow up would be interesting also of how parents cope with the high rate of false positives with RAA and CNVs” Lane 378-379.

Reviewer 2 Report

this methodology should be called as NIPS (non invasive prenatal screening), as NIPT “testing” is confused as a diagnostic test.  This is supposed to be a screen, which, if positive,  needs a confirmatory test.

-          Introduction should also cite the systematic review and the evidence based guideline by ACMG

o   Systematic evidence-based review: The application of noninvasive prenatal screening using cell-free DNA in general-risk pregnancies PMID: 35608568

o   Noninvasive prenatal screening (NIPS) for fetal chromosome abnormalities in a general-risk population: An evidence-based clinical guideline of the American College of Medical Genetics and Genomics (ACMG) PMID: 36524989

-          Line 149 – if one of ten has positive NIPS result, wouldn’t it be “remaining NINE samples”?

-          Line 191 – it says 17 samples found to be high risk for T21. But table 2 says 18.

-          Line 205 – 1/1110 was NIPS negative, but it was really a true positive?  Any more information on this patient/sample?

-          Line 221 – 222: 1110 were found to be true negative. But line 205 says specificity is 1109/1110?  There is one false negative?

-          Line 240: case #5 should be trisomy 8???  Did the authors mean case #6?

-          Line 244: trisomy 8 case should be case #5.

-          Line 251: but there are four CNV mentioned in table 3?

-         

Author Response

this methodology should be called as NIPS (non invasive prenatal screening), as NIPT “testing” is confused as a diagnostic test.  This is supposed to be a screen, which, if positive,  needs a confirmatory test.

Response: Thanks for your suggestion. We agree with the reviewer that prenatal screening cffDNA based is a screening test and not a diagnostic test. We replaced NIPT with NIPS in all the manuscript.

-          Introduction should also cite the systematic review and the evidence-based guideline by ACMG

o   Systematic evidence-based review: The application of noninvasive prenatal screening using cell-free DNA in general-risk pregnancies PMID: 35608568

  • Noninvasive prenatal screening (NIPS) for fetal chromosome abnormalities in a general-risk population: An evidence-based clinical guideline of the American College of Medical Genetics and Genomics (ACMG) PMID: 36524989

  • Response: Thanks for your comment. As the reviewer suggested, we added the references with PMID 35608568 and PMID 36524989 in the Introduction section.
  •  

-          Line 149 – if one of ten has positive NIPS result, wouldn’t it be “remaining NINE samples”?

   Response: Thanks for your comment. We added the sentence: “Of these, one received a high-risk NIPS result for trisomy 21, with follow-up analysis showing that this was a true positive. The remaining nine samples received a NIPS result of ‘no fetal anomaly detected’ and all were confirmed as true negatives” Lane 154-157

-          Line 191 – it says 17 samples found to be high risk for T21. But table 2 says 18.

   Response: Thanks for your comment. We replaced 17 with 18 at lane 191 (now lane 197)

-          Line 205 – 1/1110 was NIPS negative, but it was really a true positive?  Any more information on this patient/sample?

-          Line 221 – 222: 1110 were found to be true negative. But line 205 says specificity is 1109/1110?  There is one false negative?

Response: Thanks for your comment. As the reviewer suggested we checked the sensitivity and specificity values. Specificity is TN/(TN+FP), so the calculation was 1109/1110, because we have one discordant case for trisomy 21.

-          Line 240: case #5 should be trisomy 8???  Did the authors mean case #6?

Response: Thanks for your comment. As the reviewer suggested we replaced case 5 with case 6

-          Line 244: trisomy 8 case should be case #5.

Response: Thanks for your comment. As the reviewer suggested we replaced case 4 with case 5

-          Line 251: but there are four CNV mentioned in table 3?

Response: Thanks for your comment. As the reviewer suggested we replaced three CNV with four CNV.